# Power and/or Penury of Visualizations: Some Thoughts on Remote Sensing Data and Products in Archaeology

## Włodzimierz Rączkowski 

Faculty of Archaeology, Adam Mickiewicz University in Poznań, 61-614 Poznań, Poland; wlodekra@amu.edu.pl; Tel.: +48-61-829-1421

**Abstract:** Airborne and spaceborne remote sensing in archaeology generates at least two important issues for discussion: technology and visualization. Technology seems to open new cognitive perspectives for archaeology and keeps researchers increasingly fascinated in its capabilities (archaeological science being a case in point). Acquired data, especially via remote sensing methods, can be studied after processing and visualizing. The paper raises several issues related to the new cognitive situation of archaeologists facing the development of new technologies within remote sensing methods. These issues are discussed from ontological, epistemological, and discursive perspectives, supporting an exploration of the role of technology and visualization. The ontological perspective places the visualization of remote sensing data in the context of understanding Virtual Reality and Jean Baudrillard's simulacra. The epistemological perspective generates questions related to visualization as *mimesis*, the issue of cultural neutrality, and the use of sophisticated classifications and analytical techniques. The level of discursiveness of visualization includes categories such as persuasion, standardization, and aesthetics. This discussion is framed in relation to Martin Heidegger's understanding of technology and a dichotomy of naturalism versus antinaturalism.

**Keywords:** Martin Heidegger; technology; visualization; mimesis; remote sensing archaeology; cultural context

## 1. Introduction

Image is one of the most basic attributes of archaeology. Imagery has long been present in archaeology as a way of documenting/confirming the presence of relics of the past, thus serving to communicate about the past and creating visions of the past (e.g., [1]). Over time imagery became part of archaeological analysis. The emergence of photography in the mid-19th century created a new cognitive situation in the development of the understanding of the world. It was photography that had the ability to reproduce the world [2]. Photography and other techniques of visualization of data and imagery of the past have slowly entered archaeology, and in recent years this process has accelerated rapidly. Dave Cowley [3] (p. 18) noted that the application of new technologies in archaeology involves "[ . . . ] often expressions of shifting fashions in archaeological practice. There is thus a danger that changing fashion, rather than intellectual rigor, will heavily influence the application of particular methods, often with little of understanding of underlying principles." I fully agree with this view because I have witnessed the phenomenon of almost universal introduction of new technologies (especially remote sensing methods) in archaeology without sufficient critical reflection. Cowley [3] (p. 18) constructs his deliberations within the framework of dichotomies: "believers vs. non-believers," "traditional vs. progressive," or "old vs. new." I would like to approach the problem differently, without deciding what is better or worse, by focusing on the cognitive aspects associated with the introduction of new digital technologies to archaeology. The main axis of the

narrative that determines the place of technology in the cognitive process will be the dichotomy of naturalism and antinaturalism. During the past several decades in archaeological uses of photography (and other forms of imaging) several processes have significantly changed the cognitive position of different forms of visualizations (understood here as photographs, images, 3D models, animations etc.,). These processes are: (1) opening archaeology to philosophical reflection and the role of archaeological theory (processual archaeology, Marxist archaeology, post-processual archaeologies, symmetrical archaeology etc.); (2) *linguistic turn* (understood as the shift in philosophy emphasizing the importance of the structure and usage of language in human meaning-making and communication) in relation to photography (Roland Barthes, Susan Sontag) and, consequently, the way the image is considered as text; and (3) technological changes (especially the rapid development of remote sensing methods) regarding data acquisition, analysis, and visualization.

The relationship between image/visualization and technology is crucial in these processes. It is technology that has significantly changed the ability to generate images related to the study of the past (cf. [4]). The question arises to what extent these changes have affected the understanding of the place and the cognitive status of image/visualization in archaeological research. How can the visualization of remote sensing data be treated in the context of contemporary philosophy and how does this translate into the cognitive process in archeology?

The technology–image relationship (in the context of the naturalism vs. antinaturalism dichotomy) is key in the analysis of these problems. I would like to look at this from ontological, epistemological, and discursive perspectives (Figure 1) using the example of remote sensing data application in archaeological research. On the one hand, these perspectives generate certain specific cognitive categories, and, on the other hand, permeate each other, which means that in the narrative below repetitions, shortcuts, or leaps of thought may appear.

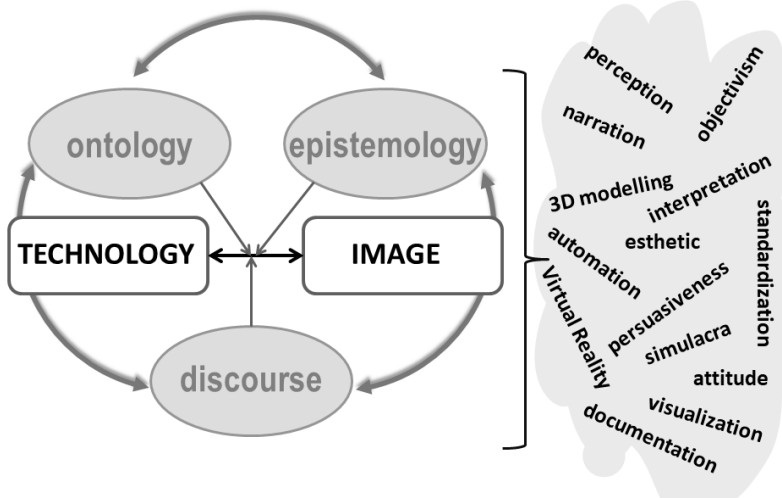

**Figure 1.** The complexity of research structure and categories in the study of technology–image/ visualization relations.

## 2. Starting Point: Identification of Research Practice Area

Contemporary archeology is a broad spectrum of research practices referring on the one hand to the tradition of culture-historical archaeology, and on the other hand, reflecting trends in contemporary philosophy and social theories (e.g., ANT, agency, symmetrical archaeology etc.,) [5,6]. Not all of these trends use remote sensing methods equally. The most common application is documenting traces of the past through various forms of visualization. The aim is to accurately describe the relics of the past so that they can be further examined. Visualizations created by remote sensing methods may also create the impression of an objective product. In a situation where documentation has a pictorial character, the principle of "what you see is what you get," i.e., objectivity and "superculturality" of

seeing, applies. Such thinking and practice are part of Plato's concept of *mimesis*, i.e., the belief that a perfect reproduction of a prototype is not a representation, but is identical with it, i.e., with the original (Figure 2). Contemporary data acquisition techniques (e.g., digital photography, airborne laser scanning (ALS), satellite imagery) enhance this view. It is therefore not surprising that visualizations meeting the criteria of photogrammetry or 3D models are treated as identical with the depicted objects, features, or landscapes.

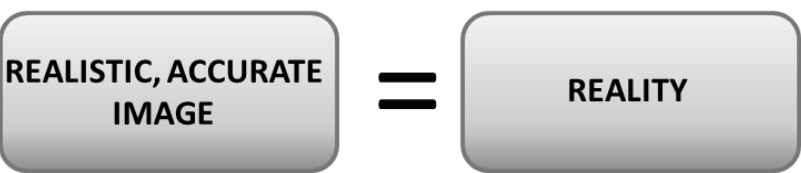

**Figure 2.** Plato's concept of *mimesis*.

Remote sensing technologies were developed within formal sciences and were adopted by archaeologists to better and more effectively study the past. The use of data visualizations obtained with remote sensing methods as objective empirical data situates this approach in archaeologies that can be classified as naturalism. The philosophical turn of the mid-20th century questioned the possibility of objective cognition and thus the dominance of the existing model of formal and natural sciences. Science has been incorporated into the area of culture, and scientific activities are conditioned by the cultural context. Therefore, all research becomes culturally constituted, and the image of the world constructed by research gains an historical and discursive dimension. Thus, the division between culture and nature was removed, and natural sciences lost their ability to present the world objectively [7]. This is a certain simplification that I accept in this text, because the problem of the relationship between nature and culture has been approached differently within the framework of contemporary philosophy (e.g., [8–10]). However, it is from this perspective that I will examine archaeological research practices using remote sensing methods and the visualizations they generate, and try to consider the issues outlined above.

In the context of *mimesis*, the space between reality and its realistic image has been filled in contemporary philosophy with the presence of language, culture, and individual experience. Additionally, the author of the representation and its recipient (Figure 3) have also appeared in this space. The whole relation has been fundamentally changed, and even the existing ontological and epistemological sense of representation has been challenged. This creates a new cognitive perspective based on inspirations developed by philosophy after the *linguistic turn* in which both the language and cultural context of actors play a crucial role, fundamentally different from the previous one (naturalism).

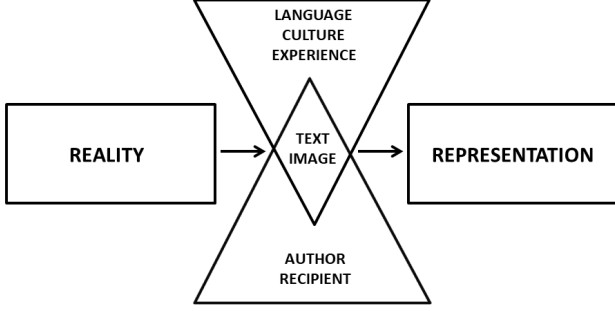

**Figure 3.** Diagram of the relationships between reality and its representation, accounting for the complexity of the space between them, inspired by postmodern philosophy.

These phenomena are also reflected in broadly understood aerial archaeology. The objectivity of aerial photography in the study of the past has been questioned and, as a result, there have studies of the interaction of aerial photography (and its interpretation) with the cultural context [11–16]. This is particularly evident in the discussion of the cognitive status of aerial reconnaissance and taking oblique aerial photographs. The cultural entanglement of the aerial photographer and the randomness of the recorded relics of the past are the part of this discussion. This observed "bias" (understood as the impact of existing knowledge, cultural prejudice, and methods, for example) was challenged to become as limited as possible in the research procedure. To some extent, it is the new technological developments that are supposed to minimize the "bias" and increase the objectivity of the research approach [17]. Technology is meant to reduce subjectivity and restore, at least to some degree, the possibility of objective knowledge (description) of reality.

## 3. Technology and Data Visualization: An Ontological Perspective

The question of technology is important in view of the considerations discussed above because it is with the help of modern technology that we acquire, process, analyze and share data. It is also commonly believed that the more technologies we use, the more objectivity in the research procedure (including data acquisition, analysis and visualization) there is. Therefore, when considering the issues of technology and visualization as its result, we must ask ourselves two questions: What is technology? and What is visualization? Both questions have significant ontological foundations.

As far as modern philosophers are concerned, it was Martin Heidegger, among others, who dealt with the place of technology in the modern world. He proposed a specific approach to technology and an understanding of the risks it entails [18,19]. For Heidegger, the essence of technology is extremely complex and its technical dimension is not the most important. Technology in his view is both "a means to an end" (instrumental definition) and "a human activity" (anthropological definition). These two approaches are closely related, intertwined and inseparable. Technology allows beings to actualize and thus leads to the revealing of reality. Therefore, it is connected with the way the truth is revealed, not only with the instrumental attainment of specific results. This understanding of technology also indicates its role in cognition. Revealing the truth in this sense is reaching the truth by a being in its being-in-the-world, and not just discovery/ recognizing the presence of something by a being (as it is often understood in traditional archaeology and reduced to "identification" of relics on an aerial photograph or in other remote sensing data). And this element is particularly important: revealing the truth with the use of technology belongs to a being, it allows it to discover the truth as it understands it. Consequently, Heidegger views technology and its role in cognition as a cultural activity. Thus, technology is not culturally neutral, although it is often perceived and treated as such (this is how naturalism puts it, and in archaeology it is particularly strongly emphasized by processual archaeology and the so-called archaeological science). For Heidegger, this culturally neutral treatment of technology is the greatest threat because it leads to the concealment of the truth. Following this thought, Marcelo Vieta and Laureano Ralon propose the term "being-in-the-technologically-mediated-world" [20] (p. 38). This emphasizes the idea that the world, things, and truth are revealed to human beings when "encountered, manipulated, or generally engaged with." Technology in these relationships becomes a "medium" that allows transmission and transforming human experience and activities [21] (pp. 56–61), [20] (p. 43). This perspective assigns a sense to the image production technology similar to the role of language in culture as it is understood after the *linguistic turn*, and it remains in contradiction with positivist/traditional thinking.

The way the image/visualization is presented in contemporary philosophy has also changed fundamentally. While in the 19th century images were still treated as representations of reality (e.g., still-life paintings), the introduction of photography exempted art from presenting the world as it is. This role was adopted by photography, and this new way of thinking about it became established. The technique of photography opened up new cognitive possibilities (e.g., the structure of movement imperceptible to the naked eye and identifiable in slow motion [22], and its neutrality was in no

way questioned (cf. [23] (pp. 27–29)). Critical reflection on photography and the role of language in its cultural presence (Roland Barthes, John Szarkowski, Susan Sontag) and other forms of visual representation led to the emergence of the terms *pictorial turn* (William J.T. Mitchell) or *iconic turn* (Gottfried Boehm) (e.g., [24,25]). This initiated the Visual Studies trend, which is a critical reaction to views emphasizing the dominance of language in culture and the study of reality as a consequence of the *linguistic turn* (e.g., [26]). In Visual Studies, culture is based on images as a primary and basic form of perception and understanding of the world. These images can be diverse and include optical or imaginative forms that present the world and its aspects in forms ranging from traditional paintings to any contemporary visualizations (including computer-generated imagery). Thus, to a large extent, these are now products of technology applications (from digital photography to computer visualizations), and their culture-forming role also applies to science (cf. [1]). Images as cultural constructs co-create reality. Therefore, the study of images and their cultural dimension should also consider the relationship between "the seen" and "the seer" [27] (references to Ferdinand de Saussure and his studies of language, and Barthes and his studies of photography).

The use of modern technology (mainly computer technology) in generating visualizations of data (including remote sensing data) related to objects to which we attribute a connection with the past, raises the question of what they are. I think we can treat them as images, i.e., created visions of objects, landscapes, and even events related to the past. If so then they constitute cultural entities, which we can put into textual categories and apply to them discourse terms introduced by Barthes with regard to photography [11,28,29]. However, such visualizations created by information technology can contain both real and fictional elements. And this significantly changes their cognitive status by opening up the possibilities of treating them as inscribed in the understanding of Virtual Reality (VR) [30–32] and by adding philosophical concepts formulated by Jean Baudrillard [33] to the cultural discourse.

VR is a technology that allows the user to create interactions with a computer-generated world regardless of its relationship with reality or imagination. Since the advent of VR technology in the 1980s, many different definitions have been formulated. Some emphasize technological aspects (computers, software, peripheral devices), while others focus on psychological issues (sensual experience, perception) (e.g., [30,34]). Among the characteristics of VR, in the ontological perspective the most important seem to be simulation, interaction, artificiality, and immersion (e.g., [30,31]). Simulation means a form of representing reality, but it cannot be treated as its copy. Interaction allows us to enter into specific relationships with virtual objects not always possible in the real world. Artificiality indicates that VR is a human creation, and thus it has nothing to do with natural processes. Finally, immersion means using solutions that raise the possibility of "sensual immersion" in this generated world.

The discussion of VR ontology is therefore an attempt to answer the question about the existence of objects in VR and their relation to reality (see also [35]). Features such as simulation and artificiality indicate that objects in VR are a form of emulation, imitation, and projection of what exists in reality. There is no doubt that such simulations are far from Plato's *mimesis*: "One must remember what the object in the virtual world actually is: nothing more than a bunch of binary numbers stored in a computer memory" ([36]; see also [37]). So the reproduction is never complete (i.e., the same as the original). It is limited to certain physical elements, because in VR it is not possible to visualize all their complex properties (cf. [38,39]).

In the case of archaeology, visualizations that are part of a VR can have at least three characteristics: (1) They are "models" of real existing objects that have been assigned a link to the past, e.g., an axe, a fibula, an amphora; (2) they are images of space/landscape in which there are elements considered to have been created by humans in the past (e.g., Stonehenge area, Giza pyramid complex—see [40–42]); and (3) they are situations and fictional phenomena (e.g., images of events, people's economic and social activities, or religious rituals) (e.g., [43] (pp. 94–96)). If (non-)real objects are simulations, forms of projection or imagining reality, this invites the conclusion that they are, in consequence, products of the

human mind. This means that they are not only the products of information technology, but also—and perhaps even, in particular—of human reflection on the world [30] (pp. 148–149).

Considering the reflection that objects in VR imitate or simulate real objects, it becomes obvious that references to Baudrillard's proposal [33] should be made. According to him, simulation means pretending, illusion, creating a world of imagination, and the products of this process, i.e., simulacra, pretend to be objects from reality, which are widely/commonly treated as real objects from the past. This means that visualization imitates objects but do not present them. Therefore, one can probably formulate a conclusion that visualizations of data (including remote sensing data) in archaeology constitute an imagined, modelled world, materialized by means of computer technologies. In this sequence of technological connections there are devices allowing for data acquisition (products of the human mind), computer software that processes them (a programmer's product), and processors (also constructed by humans). Visualizations may be revised, changed, or modified at each stage of this process and in consequence even create new visualizations of the same real objects (cf. [36]). From this perspective, visualizations of any form and any data are not ontologically significantly different from traditional photography, especially if we agree with Szarkowski's view that: "Photography is a system of visual editing. At bottom, it is a matter of surrounding with a frame a portion of one's cone of vision, while standing in the right place at the right time. Like chess, or writing, it is a matter of choosing from among given possibilities, but in the case of photography the number of possibilities is not finite but infinite" (quotation after [29] (p. 150)).

## 4. Visualized Data and Cognitive Processes (Epistemological Perspective)

If ontological reflection gives us an idea of what visualizations in archaeology are (e.g., as a result of remote sensing methods application), then we should consider the connection between visualized data and the cognitive process. Again, recalling Heidegger [19], one may ask the following question: Does contemporary technology allow the world to reveal itself as it is? In terms of archaeology this question may be: Does technology allow the past to be revealed? In relation to ontological findings, the following question can be asked: What cognitive function does a simulation model/image/visualization play in the process of building knowledge about the past? Heidegger emphasized that technology was oriented toward revealing the truth as offered by natural sciences [44]. Thus, the reference to the role of technology in the cognitive processes in archaeology places the researcher in the naturalistic trend [45]. In the dominant archaeological research practices (cultural-historical archaeology, processual archaeology) there is no cognitive dissonance here, because they fit into this trend. This has far-reaching consequences in terms of formulating research questions and methods of solving problems. One of the traits of thinking in the spirit of naturalism is the recognition of the neutrality of the aforementioned technology (e.g., [46]). This means that in the process of revealing, it is possible to obtain and analyze data objectively, which only at the final stage of the research process is subjected to evaluation/interpretation (and even very often confined only to an objective description and presentation of results). In this understanding, technologies that allow visualization of data obtained in a culturally neutral way lead to learning about the past world. In this cognitive perspective, such an image/visualization, being a faithful representation of reality (e.g., documentation) in the Platonic understanding of *mimesis*, becomes the basis for building knowledge about the past (Figure 2), providing useful knowledge about the past. Thus, in this approach, the image/visualization constitutes an objective, empirical record allowing the formulation of valid conclusions and judgments about the past.

An anti-naturalistic perspective significantly changes the cognitive status of image/visualization. Barthes's reflections on photography questioned this way of thinking about images in whose creation an important role was played by technology (cf. also later reflections by Bruno Latour [47] (pp. 111–120). The seeming absence of a human in the process of creating an image in no way guarantees the objectivity of reality representation. Barthes (e.g., [28,48,49]) reflected critically on photography (still analog then) in two respects: phenomenological and hermeneutical. Both significantly change the way of thinking about photography as a representation of reality, and entangle photography into complex

social relations, the cultural role of language, and perceptual abilities (e.g., [50,51]). Can this thinking be applied to the analysis of the role of visualizations produced by techniques other than the camera (e.g., ALS, Image Based Modelling, satellite imagery, geophysical methods) in the process of learning of the past?

In the application process of all remote sensing methods currently using new technologies, it is possible to distinguish the stages of planning and preparation of measurements, data acquisition, data processing, analysis, and final visualization. In a naturalistic, positivist approach, only the final visualization, as identical to the original, becomes the subject of research. However, the approach proposed by Barthes and other postmodernist thinkers indicates that at each of the mentioned stages of research procedure we are dealing with individual decisions, whether of an expert in data acquisition and processing, or of an archaeologist formulating his or her expectations and evaluating the final product (e.g., [52–54]).

## 5. Data Acquisition

Technology has opened up new possibilities of data acquisition, the purpose of which is to provide information about reality. This is relevant to remote sensing methods and their applications in archaeology. It is commonly believed that these methods allow collecting information on terrain features associated with human activity in the past (aerial photographs, ALS, image-based modelling and rendering, satellite imagery, etc.,) or relics under the surface of the earth (aerial photographs, optical satellite imagery, geophysical methods, etc.,). Each method and each device records the specific characteristics of features and landscapes. It is also not the case that all data-collecting sensors possess identical characteristics that were apparent and within the sensing capacity of the device at the time of recording (e.g., [55]). This is illustrated by a comparison of spectral bands of selected optical satellite sensors (Table 1).

**Table 1.** Comparison of spectral bands of selected optical satellite sensors.

|  | Blue (μm) | Green (μm) | Red (μm) | NIR (μm) |
|---|---|---|---|---|
| **IKONOS** | 0.45–0.52 | 0.52–0.60 | 0.63–0.69 | 0.69–0.90 |
| **GeoEye-1** | 0.45–0.51 | 0.51–0.58 | 0.655–0.69 | 0.78–0.92 |
| **OrbView-3** | 0.45–0.52 | 0.52–0.60 | 0.625–0.695 | 0.76–0.90 |
| **QuickBird-2** | 0.45–0.52 | 0.52–0.60 | 0.63–0.69 | 0.76–0.90 |
| **WorldView-2, 3** | 0.45–0.51 | 0.51–0.58 | 0.63–0.69 | 0.77–0.895 |
| **Pleiades-1A, 1B** | 0.43–0.55 | 0.49–0.61 | 0.60–0.72 | 0.75–0.95 |
| **SPOT 7** | 0.455–0.525 | 0.53–0.59 | 0.625–0.695 | 0.76–0.89 |

As a consequence, the data record different (even if only minimally) characteristics of objects that we later assign (or not) a connection with the past. The same applies to cameras (image sensors, lenses), recorders of different wavelengths (hyperspectral and multispectral sensors), devices used in geophysical methods, or airborne laser scanning. Device designers respond to the formulated needs, apply solutions according to their knowledge, and this means they are not culturally neutral in Martin Heidegger's understanding. As a consequence, the acquired data cannot be treated in a culturally neutral way either. In addition to the devices that represent the technological domain, there are also experts performing tasks assigned to them and/or archaeologists who formulate (precisely or not) their expectations. All this arrangement of relationships affects data retrieval and final format (Figure 4). Their relationship to reality is culturally shaped in many dimensions that are not entirely controllable and deconstructible. Evaluation of data conformity with reality (e.g., evaluation of measurement or scale precision), providing that it is possible at all, is affected by the views and in accordance with parameters and standards adopted by a given scientific community (e.g., [56]). Therefore, data gathered by means of various techniques are culturally oriented in their nature, due to the purpose, choice of devices, or method used.

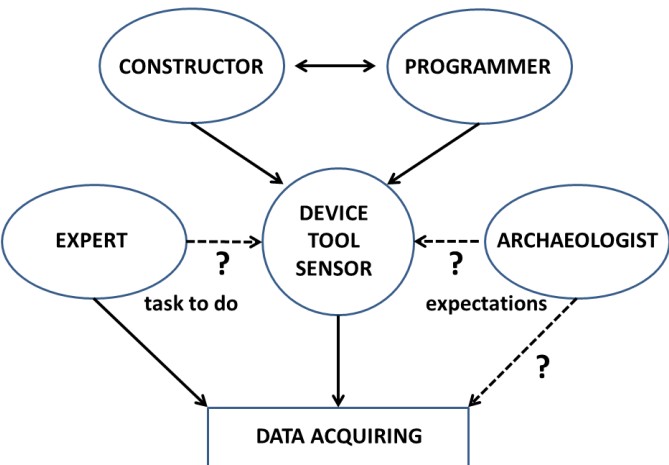

**Figure 4.** Cultural arrangement of relations within the process of data acquisition in archaeology.

## 6. Data Processing and Visualization

Culturally acquired digital data are subject to complex processing (nowadays almost exclusively computer-based). On the one hand, there are specially designed computer programs (the role of programmers) or existing software (also the work of programmers). On the other hand, data processing (evaluation of data quality, application of various algorithms, visualization of semi-finished products and products) relates to the formulated expectations of archaeologists. This process can be long-lasting, and various solutions can be applied until the image is generated in accordance with the imagination of the recipient (here most often the archaeologist). The inability to create such an image/visualization acceptable to the archaeologist often leads to rejection of data, discrediting the method/device, or even questioning the competence of the expert involved.

Is the archaeologist, as the first recipient of the final visualization, aware of the technical complexity of the data acquisition and processing processes (Figure 5) and, consequently, of the degree of reduction, manipulation and modification that occur throughout them? Between the formulation of the task and the final visualization there are at least two *black boxes* (in the understanding of Latour [57] (p. 304)) representing the areas of intensive use of technology, in which processes and manipulations usually take place uncontrolled by the archaeologist, but usually recognized by remote sensing methods experts. If the archaeologist is unaware of these processes, conditions arise for formulating an opinion about the cultural neutrality of technology and, consequently, visualizations. Hence, it is only one step to recognize that visualization is identical with reality (*mimesis*) and does not generate epistemological doubts.

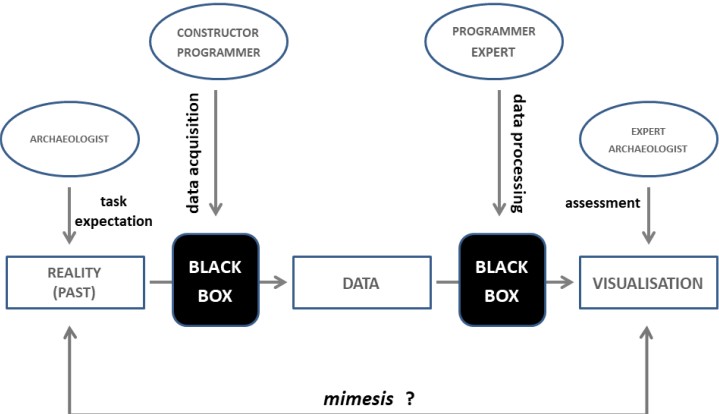

**Figure 5.** The place of technology in the process of creating visualization as a form of knowledge about the past.

Visualization is therefore a construct that is the result of many complex decision-making processes based on knowledge, technical capabilities, skills, goals, expectations, and awareness of research proceedings. Therefore, at each stage of the research process, signs, i.e., sets of things (including images), are generated, which according to the accepted cultural rules should evoke a specific thought or association in the recipient [58]. The data presented in the form of signs are subject to interpretation. Following Baudrillard, these are only imitations, not replications of reality. In this context, I can return to the question: Can these simulacra be the basis for (re-)constructing the past, playing some role in the process of learning about the world? In Baudrillard's view, visualization is not a presentation or an illustration of events (also past ones), because in the era of total simulation it is not possible (cf. [59]). Creating visualizations is, therefore, not a representation of an object from the past, but a cultural event in itself, here and now. An electronically generated visualization can be processed infinitely by creating new simulacra. Seemingly, such visualizations suggest realness, thus threatening the differences between the real and the imaginary, and between the true and the false [60].

By relating visualizations to the cognitive processes, a certain perspective begins to emerge, probably far from common thinking. In the research procedure, two levels on which the creation of meanings takes place become more and more clearly visible. Technology is connected with revealing, which is inevitably associated with assigning meanings to objects, even if only by linking them to the past. In data, these objects are often not readable directly, but only through data processing, and there is no doubt that this level is conditioned by the cultural experience of an archaeologist (or a data expert, not necessarily an expert archaeologist). The visualizations of these interpreted features/sites/landscapes, or rather their simulations, lead to the subsequent assignment of meanings (Roland Barthes, Jean Baudrillard). As a consequence, the simulacra archaeologists/ experts generate obscure reality, and they are not aware of it. This refers to the threats resulting from the use of technology that Heidegger wrote about. Archaeology becomes as virtual as the world around it. Archaeologists stop investigating the past and they only examine its images. Visualizations of data do not reflect reality (and even more so the past), but introduce a new system of signhood, which is subject to cultural interpretation.

As a result, visualizations (and the technology allowing their production) are part of the process of giving meaning. They are the Medium, and they become the Message [61] subject to hermeneutical interpretation (see [20] (p. 43)). This understanding opens up the possibility of looking at the hermeneutical potential of image technology from the perspective of Marshall McLuhan's tetradic analysis [62,63]. This analysis allows us to assess technology (McLuhan's laws of the tetrad) as a shared method. Marshall McLuhan's Heideggerian-inspired human–technology relation, on the one hand, expands certain forms of human activity and, on the other, reduces/obsolesces them [20] (p. 52). This means that teradic analysis can be treated as a tool (probably not the only one) for a more in-depth understanding of the cultural functioning of the meanings given to visualizations, but also as building a conceptual bridge between the epistemological and discursive perspectives (visualization as a medium, message). The tetradic model of these relations involves: (1) intensification/enhancement of some aspects of individuals or social groups' activities within a given culture; (2) obsolescence of some capacities of people or social groups; (3) retrieval of something from a previous activity or capacity; and 4) reversal of something into its opposite, when taken to its limit (Figure 6).

Depending on the archaeologist's cultural context, experience, knowledge, and expectations, the same aspects of the archaeologist–technology relationship (and visualization) will be dealt with differently within the tetrad. Consequently, the cognitive value of visualization will be culturally constructed and will have a historical dimension. Archaeologists have dealt with such a phenomenon in the context of the use of aerial photographs in archaeology. The perception of their cognitive value (but also the way of taking photos) changed depending on the needs of archaeologists, resulting from the emergence of new theoretical concepts and research questions [12,15,51] and this phenomenon was relatively well recognized recently. The emergence of new trends did not necessarily replace the earlier

approaches, which meant that, to varying degrees, there existed and do exist simultaneously various practices assigning different cognitive value to aerial photographs.

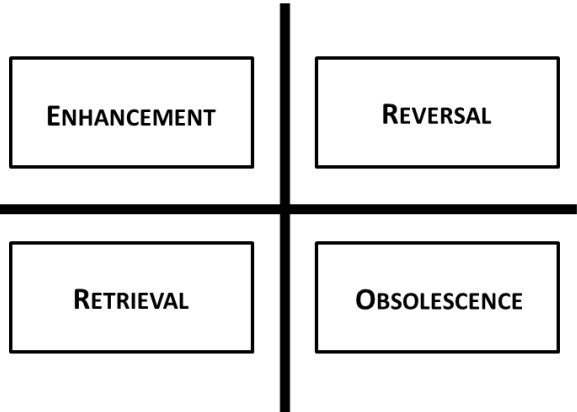

**Figure 6.** Diagram of Marshall McLuhan's tetradic model.

The historical dimension of the cognitive value of aerial photographs/remote sensing data relates not only to the methods of obtaining them, but to their processing leading to the culturally expected results. This is what archaeology is facing now in connection with new methods of obtaining mass data. Currently, this factor encourages an emphasis on technical solutions with limited critical reflection on the cognitive value of remote sensing data and generated visualizations, as was the case with aerial photography in archaeology.

In the context of the cognitive value of data, it is worthwhile to take up the subject of the significant change resulting from the introduction of new methods of data acquisition. Methods such as ALS and satellite imagery provide so much data that traditional, manual, and visual methods of data processing, analysis, and interpretation are no longer effective [3]. In archaeological studies of small areas or sites, this is not a serious problem, but in landscape studies or in building strategies for managing archaeological heritage over large areas it has already been a challenge. It is therefore not surprising that since the 2010s discussion on the use of (semi-)automatic feature recognition or (semi-)automatic detection in remote sensing archaeology (see [64,65]) has begun. This is probably an unavoidable process and archaeologists should face it conceptually [66,67].

In current archaeological practices, three basic strategies are being developed, which are, in fact, procedures of data classification: pixel-based classification, object-based image analysis (OBIA), and machine learning (ML) implemented with the use of modern tools such as neural networks (e.g., [65,67–70]). In the first strategy, pixels are the basic classification units. Different algorithms are used to classify data, and the strategy is optimal if it operates with well-defined and well-separated classes of pixels corresponding to unique archaeological/landscape features. This is not always possible and can lead to an incorrect classification of a pixel. The second strategy is an attempt to circumvent these limitations. OBIA methods use image segmentation into identified homogeneous objects using multiple variables (pixel value, shape, texture, geographic components, etc.,) [65,67,69]. Both strategies are supervised methods of classification. The third strategy, i.e., ML, has recently been added to the range of strategies for classifying archaeological objects. ML is a set of algorithms for recognizing patterns (e.g., support vector machines—SVM, artificial neural networks—ANNs, Random Forests—RFs, $k$-nearest neighbor—$k$-NN, etc.,) that allows (1) modelling complex class signatures, (2) accepting a variety of input predictor data, and (3) not making assumptions about data distribution [71]. Currently, ML also includes deep learning (DL) strategies, which are techniques permitting the construction of a multi-level analysis and classification architecture. In particular, this concerns the improvement of relations between false-negative and false-positive results owing to the introduction of several levels between the input layer and the output layer (e.g., [72,73]). Within DL

different methods can be used as needed, e.g., recurrent neural networks—RNNs, or convolutional neural networks—CNNs.

Regardless of the differences arising from the nature of data and the classification algorithms used in (semi-)automatic detection procedures, a number of essential stages can be distinguished in these strategies. The first is data acquisition and pre-processing. At present this strategy most often involves ALS data, allowing preparation of DTM and its multiple visualizations (often trend removal techniques such as LRM), and satellite imagery (optical and/or SAR). All pre-trained datasets expect input images to be normalized. The next step is the development of a dataset as a reference for the classification process. This dataset, depending on the strategy adopted, can take the form of defined groups of pixels or a set of object images as templates (based on image segmentation). In practice, existing resources are often used (e.g., ResNet, GoogLeNet, ImageNet), or a custom set of images is introduced which can be used as a training set (extraction of training images, augmentation and cropping). This is often applied in archaeological projects [70,72,74]. Based on the reference data prepared in this way, an automatic classification process is launched, which leads to the identification of objects that meet the adopted criteria. The output is subject to an accuracy assessment in terms of compliance with archaeologists' expectations/knowledge. If OBIA or ML is used the output is not subject to further classification procedures. For DL this process may be repeated by indicating false-negatives and false-positives. In this way, the algorithm "learns" to recognize the expected objects. One training round (epoch) is included in the subsequent cycles. The end of the process depends on the decision of the archaeologist/expert [74]. The final stage is validation of the results, which may involve a desktop survey and/or fieldwork (different versions of ground truthing) [65,70,72]. From a purely technological perspective this seems to be a process that eliminates the weaknesses of a traditional interpreter-based approach. In practice, from an epistemological perspective, it is not very different from the latter. In both approaches the interpreter's knowledge, expectations and experience are decisive.

Many authors emphasize "expert knowledge-driven" decisions, which relate to all the aforementioned stages. The quality of the prepared output data significantly affects all subsequent stages [67] (p. 487), including image segmentation (OBIA) and creation of training datasets [69] (p. 2). In DL the possibility of the cycle repetition is left to the expert (archaeologist?) to decide on the completion of the process. This decision, and the evaluation of results in this classification model (also in others), depends on the archaeologist's expectations. The question is if the network can overlearn, and how this may affect the information obtained. Is it possible to identify such a problem? What could be the consequences for the archaeologist's expected outcome? Is the archaeologist aware of the consequences or possible need for data modification?

I have no doubt that all these stages are consecutive steps in data work comprising certain communicative arrangements that involve making cultural decisions, creating signs, and assigning meanings. Logical consistency, which is to guarantee the construction of algorithms allowing for a quick and "objectivized" data search, is based on our expectations, knowledge, and previous experience.

The assignment of meanings is involved at the all stages of these classification strategies. The attempts made so far have concerned selected categories/classes of archaeological sites (mounds, barrows, roundhouses, charcoal kilns, Celtic fields etc.—[72,74,75], i.e., they are based on existing knowledge, experience, and expectations. The question can be posed whether these methods allow for the qualitative identification of other/new types of relics of the past. In this context, Rodney Brooks's opinion that "[ . . . ] deep learning does not help in giving a machine 'intent' or any overarching goals or 'wants'" may be relevant [76] (p. 280).

This question indicates the need to look at the algorithms used in ML and DL. Does repetition of the process lead to such a type of "self-learning" in which objects that are radically different from those included in training data sets can be identified, i.e., beyond existing knowledge? Do archaeologists fully understand how the algorithms operate? Do they have an insight into what the network knows? If they are not able to answer these questions positively, by definition they are dealing with *black box(es)* (for both experts and archaeologists) (e.g., [77]). It also means that archaeologists are unable to answer

the question about the relationship between the obtained result and what is commonly called reality. Archaeologists can therefore assume that even despite the process of "self-learning," there is a reduction in the potential and resources of past relics contained in the data. Such are the limitations resulting from the use of a specific algorithm (problem-oriented algorithm) and its positive result. In this sense an algorithm used in detection can be treated in the same way as previous classifications in archaeology conditioned by theories, trends, and research questions (cf. [78]). In every situation archaeologists lose the diversity of the world as it is etic approach (instead of emic). And, it should be noted, using traditional methods suffers from the same problems.

A number of researchers [3,70,72,79] have expressed the opinion that the results of (semi-)automatic detection methods should be verified. It is difficult to argue with this view. But what does it mean in a cognitive perspective? In my opinion it means that the result of the algorithm(s) used is not to be considered a final determination. The final determining authority is the archaeologist, and (very often) involves some form of ground truthing (see [80] (pp. 518–521), such as field survey, digging test trenches etc., (e.g., [70,72,75]) which is an essential element of the whole cognitive procedure. I have no doubt that the perceptive and interpretative capabilities of an archaeologist are significantly different from what can be stored in the algorithm(s). Moreover, I believe that it is probably impossible to capture in mathematical equations the entire cultural complexity of the world in the past, formation processes, and the interpretative experiences and expectations of archaeologists.

The need to verify (semi-)automatic detection results accounts for only one side of this complex process. (Semi-)automatic detection places the objects selected by the archaeologist (and thus already known to us) and learned by the algorithm in our area of interest. Verification allows us to accept or reject what the archaeological/non-archaeological algorithm suggests. In this respect, the process of reduction and modification of information by the archaeologist takes place twice: first, at the stage of preparation of the input dataset, and then at the stage of verification, i.e., output assessment. Both stages lead to the rejection and reduction of world complexity. In this sense, technology in Heidegger's sense obscures some aspects of the world (including the past world) and, consequently, it (this past world) disappears in its complexity from the cognitive process. At the first stage, it is due to the fact that the input is prepared according to the archaeologist's knowledge and expectations. This stage is consequently burdened with the archaeologist's bias. As a consequence, the algorithm(s) can at best identify the objects archaeologists expect. An algorithm will magnify any patterns in the input data. It means that if bias is present at the stage of data input preparation (let alone the archaeologist's bias at the pre-processing stage) the algorithm will also magnify that bias [81]. Consequently, algorithms reflect the biases of both programmers and datasets prepared by archaeologists. It should also be noted, however, that the algorithm may point to a potential object that the archaeologist might omit in their interpretation (e.g., the question of noticing minimal alternations).

Our knowledge and experience play a role in the data assessment and verification stages leading to the acceptance or rejection of selected objects. The Archaeologist's judgment is necessary to assess the accuracy of algorithmic output [81], though the processes by which that may happen can be opaque. It is definitely a reduction of the potential and stock of past relics present in the data. Such are the limitations resulting from the use of a specific algorithm (problem-oriented algorithm) and its positive result. There is also a question about the possibility of validating its (their) functioning. At these stages of automatic detection (including DL) archaeologists focus on what the algorithm offers (even if it is finally rejected). In the "cognitive non-existence" there remain all those objects that do not meet the criteria of the algorithm and fall out of the field of the archaeologist's perception. Although they are still in the raw data, the question remains, will there be new data search algorithms, and will the new algorithms allow for a significantly similar or a different result?

At one point, archaeologists became aware of the presence of bias in their work (e.g., [14,82–84]) and almost simultaneously, initiatives were undertaken to eliminate this phenomenon (e.g., discussion on bias of oblique and vertical aerial photographs). The need to reduce bias in archaeological research procedures is often an argument for the application of high-tech methods and devices in remote sensing

(e.g., [17,85,86]). In particular, this occurs in the context of working with remote sensing data. The above considerations of automatic detection probably sufficiently demonstrate that such solutions do not lead to bias reduction. Bias permeates the whole complex classification procedure and perhaps because of that it is so difficult to grasp. However, there is also a growing awareness of its presence and inevitability (e.g., [72]), even when using sophisticated technologies.

My primary objective in considering (semi-)automatic detection in archaeology is not to assess its effectiveness in comparison with traditional methods of identifying archaeological objects. Rather, I would like to ask the following question: Does this new technology change anything in archaeologists' cognitive process, or does it modify the current thinking about the past? The existing applications of (semi-)automatic detection in remote sensing archaeology clearly demonstrate the risks mentioned by Heidegger.

For many authors, the main goal is to identify archaeological resources in areas that have so far been poorly explored. The effectiveness of the method is assessed on the basis of the extent to which it generates an increase in the number of database records (e.g., more round barrows, charcoal kilns or round houses). Archaeologists frequently are driven by their desire to find/reveal new (although already known types) sites. The question arises whether increasing the amount of empirical data (records in the database) can transform archaeology and the archaeologist's ability to explore the past? In archaeology, technology only serves to increase the number of discoveries. Conceptually, archaeologists remain at the same stage when aerial archaeology was criticized of a "stamp collection" approach (e.g., [87–89]). The fetish of empirical data (efficiency of identifying new objects) leads to a focus on discovery in the common sense (see above). And in this sense, technology is a threat to science (as understood by Heidegger). The effectiveness of technology obscures the complexity of the world, including the past world (and not only because archaeologists discover what they know), but also the complexity of research practices. This leads to impoverished reflection on what archaeologists do and how they do it.

## 7. Archaeologist as User: From Perception to Interpretation

Image technology (and the image itself) is a medium that allows the transmission, translation, storage, and transformation of a message. But it is not created without prior perception and interpretation. While hermeneutical interpretation has been present for quite a long time in humanistic discourse (Edmund Husserl, Paul Ricoeur), the reflection on perception and its cultural entanglement is not at all obvious. The process of perception is related to the inflow of external stimuli (here usually visual), which are subject to various further transformations leading to the recognition of the object (e.g., [90] (p. 6), [91]). Perception and its role in cognitive processes is now largely a subject of cognitive psychology ([92–94]), but philosophical reflection on the relationship between perception and cognition goes back to Immanuel Kant's considerations. Among many studies dealing with this issue from an epistemological perspective, it is worth recalling the works of Ludwik Fleck from the 1930s [95], which appear more and more frequently in contemporary discussions (e.g., [96,97]). In the discourse on the development of science and technology, Fleck's views place him conceptually (not chronologically) between Thomas Kuhn and Bruno Latour [97,98]. While the former, for example, emphasizes the commitment to the paradigm in science, Fleck places scientific activity in the cultural context of the researcher, i.e., in line with the anti-naturalistic approach proposed in this paper. Considering the questions raised, Fleck's assertion that "To see is to know" seems fundamental. Certainly, such a statement does not raise any fundamental doubts today, but in the period when it was formulated it must have had a very controversial resonance. With this assertion Fleck questioned perception as a means of objective learning about the world, and thus the foundations of science built on Enlightenment concepts [99]. According to him, it is impossible to see without anticipating knowledge, and consequently we see what is mental and not physical (!). It is not stimuli that decide what we see, but the processes taking place in our minds. Such assertions contradict the common opinion that physical vision is objective. Ludwig Wittgenstein (late Wittgenstein) [100] expressed himself in a

similar spirit, claiming that we see only an insignificant tangle of shapes and colors, which we give meaning to, but which can change. This is related to targeted or directed perception (attention, attitude, aspect) (e.g., [101]) and, in consequence, to interpretation. The way of perceiving is nothing more than a certain interest in the surroundings, but also an image/visualization. In the case of remote sensing data, the problem is even more complex and multilevel, because we observe a mediated reality in the data. In our interpretation, we usually ignore the whole process of data acquisition, processing, and generation of various forms of visualization and focus on the final product (final visualization). The perception processes are present in the data workflow much earlier. The way we see is the social arrangements (contracts) of what we see. They are so mentally close to us that we do not even notice that we are learning to see [99]. This social creation of ways of seeing illustrates very well the difference in the presentation and reading of topography in the tradition of British archaeology and other countries (e.g., [102–104]). This social dimension of knowledge-based perception was also stressed by Fleck. His concepts of "thought style" and "thought collectives" have been widely discussed in literature [96–98,105,106]. An independent (socially excluded) researcher does not exist. Everyone is entangled in certain communities, including groups of scientists with their exchanges of ideas or intellectual interactions. Itrected perception, with corresponding mental and objective assimilation of what has been so perceived. It is characterized by common features in the problems of interest to a tht. The "thought collective" is "[ . . . ] a community of persons mutually exchanging or maintaining intellectual interaction, we will find by implication that it also provides the special "carrier" for the historical development of any field of thought, as well as for the given stock of knowledge and level of culture" [95] (p. 39). The opportunity to be part of such a community is given to those who, in the course of their education, have mastered and accepted the rules in force [107] (p. 198). The "thought style" is understood by Fleck [95] (p. 99) as "[the readiness for] directed perception, with corresponding mental and objective assimilation of what has been so perceived. It is characterized by common features in the problems of interest to a thought collective, by the judgment which the thought collective considers evident, and by the methods which it applies as a means of cognition. The thought style may also be accompanied by a technical and literary style characteristic of the given system of knowledge."

In science, the consequences of "thought styles" and "thought collectives" are far-reaching. Of course, they also concern archaeology and the use of visualization in the study of the past. Belonging to "thought collectives" perpetuates the belief in the rightness of the adopted views and research conduct. As a consequence, scientists have identical or very similar views in a given field. They do not realize the nature of their own thinking, conditioned by the socio-historical context. The conviction is formed that since everyone thinks the same, it is the only possible culture-independent way of seeing the world [107]. The dissemination of certain views in society (outside the research community) co-creates common-sense knowledge that is not even subject to verification. This is in line with Heidegger's concept of concealing the world by formulated and accepted views, and with the lack of critical reflection on such views resulting in "idle talk." In the context of archaeological development, such deeply rooted and unconscious convictions are "belief in experience/observation," "objectivity of facts," "pure description and classification," and the possibility of "knowing the past." They originate from the 18th- and 19th-century concepts influencing science (empiricism, positivism, evolutionism) and are particularly strongly rooted in cultural-historical archaeology.

According to Fleck, two elements in the most popular views, in particular, affect the consolidation of our beliefs about the objectivity of our findings on the (past) world. These are "technical terms" and "scientific device." The specific power of scientific terms consists, to a large extent, in detaching their significance from the subject of cognition, hence in establishing the "objective" meaning. In this way the object being defined becomes independent, as if possessing absolute existence [108] (p. 108). Even more so, the cultural dimension of data acquisition for visualizing the (past) world resounds in the context of the "scientific device." "The analysis of epistemological significance of a scientific device would also require a separate study. It can be mentioned briefly that a scientific appliance, which is realization of

some result of a definite thought-style, directs our thinking automatically on to the tracks of that style. Measuring instruments force one to apply the notion of unit for which they were constructed; even more so, they force one to apply the notion from which they originated [...]" [108] (p. 109). From this perspective, there is no gap between the process of cognition in Fleck's viewpoint and the views of contemporary philosophers significant in the context of the use of images/visualizations (e.g., Roland Barthes, Jean Baudrillard).

Both "thought styles" and "thought collectives" affect the perception and interpretation of the world, and their alterations modify ways of thinking about it. There is no single "thought style" or "thought collective," which means that particular groups of scientists compete against each other, as if participating in specific social games [109] within the space between the legitimate and the illegitimate image of the world. "Pure facts" do not play a decisive role in these games because facts do not exist independently of cognition, and this cognition refers only to what is useful to the "learner" [110] (p. 267). The emphasis is therefore placed on language games, because it is through language that we communicate our views, ideas, and manipulate the image of the world. Therefore, it is important to construct a narrative in such a way that it becomes more persuasive and, as a result, it can become established and, relatively, universally accepted. This invites reflection on the discursive dimension of visualizations produced from remote sensing data.

## 8. Discursive Dimension of Visualization

In contemporary humanistic reflection, the concept of truth is far from the common understanding of it going back to ancient Greek philosophy (e.g., [111]). This also applies to the role of empirical data in knowing about the world and the determinants of knowledge legitimacy (e.g., [112]). In a fundamental way, the understanding of truth has been shifted from the external world, as a point of reference, to the knowledgeable subject entangled in culture. This means that seemingly mimetic visualizations of data are in practice involved in a variety of discourses and cultural games that include dialogue, communication, and power experience. Discursive practices permeate all spheres of culture and are part of the processes of producing images of the world (including archaeological construction of knowledge about the past). Thus two aspects that are increasingly being raised in the context of images/visualizations and their presence in the scientific/political.

Research on image persuasion is conducted, in particular, in psychology and cognitive science. Does archaeology use such knowledge and, consequently, consciously manipulate the image to achieve the intended effect in the recipient? It is difficult to find examples in literature that could support such an unequivocal assertion (although, for example, the analytical uses of DEM visualization algorithms may prove otherwise, see [113]). I think that rather a common-sense approach to the image, i.e., what is the object/site/landscape everyone sees, is the dominant one. To a greater extent, one can count on complying to certain standards and established customs in the archaeological community. This means that in terms of the persuasive power of the image, we use categories that fit into Fleck's concept of "thought style." It is this "thought style" that makes images/visualizations more comprehensible (and thus persuasive), if they are consistent with the knowledge and expectations of the audience [114,115]. John Casey [116] (p. 172–173), in his discussion of Wittgenstein's views, even formulates a claim that the unity of perception implies identical judgements. In archaeological practice, this persuasiveness seems to be multilevel and concerns at least two relations: data processing expert → archaeologist, and archaeologist → other archaeologists and the public.

Not every archaeologist using remote sensing data has the ability to work with it. Archaeologists are often unaware of the complexity of data acquisition processes and data processing (*black boxes*). As a consequence, in order for the data to be used in research, visualizations should be in line with an archaeologist's expectations. It is often an expert who prepares a visualization in such a way that it satisfies the archaeologist (regardless of whether it is to be used only as an illustration or for analytical and/or interpretative procedures—though simple visualization toolboxes are changing this dynamic).

Persuasion as a form of communication can influence the recipient in various ways. It is not only a matter of conveying a message, but also of exerting influence (even pressure) on the recipient (cf. [117]). In their reflections on the role of persuasion in advertising, Gerald Miller [118] and Richard Perloff [119] (pp. 19–20) stress that it can take on various functions: shaping, reinforcing, and changing. Shaping is a tendency to use persuasive messages to influence the actions and attitudes of the recipients; reinforcing is rather directed toward strengthening and deepening existing attitudes and actions; and changing is aimed at causing a significant change and establishing new, expected actions or attitudes [119]. Can archaeologists directly apply the reflection present in the discourse concerning the role of advertising in trade or persuasion in politics? Persuasion is addressed to the recipient; thus it is the broadcaster (expert, archaeologist) who formulates objectives related to the form of persuasive visualization. A question can be asked: is the role of persuasion (in the sender–recipient relationship) in scientific archaeological discourse discussed? Is the sender (archaeologist) aware of the significance of this message, or do they operate in the knowledge of how it is received? If we look at this problem from the perspective of the "thought collective," it may turn out that what is persuasive to me is also persuasive for the recipient as a representative of the same "thought collective" (no matter if it is a different archaeologist or the public Visualization is persuasive if it convinces me, and reinforces my views. The viewer might be less important, or is treated as a representative of the same "thought collective." Then, if something is persuasive for me, it will also be persuasive for the recipient, because it represents the same learned way of thinking. In consequence, we continue to function using the same cognitive models and remain exclusively in the area of persuasion as "reinforcement." There is no chance for a "changing" or "shaping" effect to take place because it requires critical reflection and attention to the way the viewer thinks. Such thinking appears in studies and practices related to the inclusion of children, youth, and local communities in the understanding of the past (e.g., [120]).

The power of persuasion is inseparably connected with aesthetics. It is assumed that aesthetics plays an important role in cognition, because it influences, among other things, the perception and understanding of described phenomena. In the traditional understanding, aesthetics is considered a philosophy of beauty and art, and stands in opposition to reason, knowledge, and science. At present, the approach to aesthetics has been re-evaluated toward the study of the aesthetic process which includes the creator (artist), the creative process, the work, the recipient, perception, and aesthetic values (e.g., [121,122]). It is clear that this contemporary approach to aesthetics is linked to the linguistic turn and the philosophy of postmodernity. Aesthetics has become a general trend that also encompasses science (including archaeology). As a result, the boundary between traditional science and truth and persuasive narratives is blurred. The problem is that in the persuasive, discursive sphere, the boundaries between reproduction, creation, and simulation are fading. There is a loss of attributes of the real world, and the phenomenon of "vanishing reality" can be observed. As a consequence, aestheticism is one of the processes that leads to the concealment of the sense of a visualized object [60]. This automatically evokes Heidegger's view that the sense of the phenomenon under investigation is concealed by the initial interpretation. If it is enhanced with aesthetic visualization, this effect is much stronger. Walter Benjamin [22] went even further, claiming that aesthetics can be used in politics to strengthen demagogic messages that mold the ideas of broad social groups. But, is this only in politics? Or, does it also apply to archaeology or messages about the past? Is it at the level of critical reflection on the past or methods of researching it and creating new models of narration, or is it at the level of inscribing oneself into existing research questions, which have been repeated for decades, and ways of solving them?

In traditional discourse visualization is a tool of power and domination. An archaeologist uses visualization to impose a vision of reality that they accept, but does not know how it is constructed (a *black box* product). In this way archaeologists value themselves and their research (e.g., [123]). It is in such an approach that the aesthetics of visualization plays a decisive role. The data itself becomes less important. The visual attractiveness of generated images or 3D models brings a fascination with the potential of technology and becomes a part of technological fetishism in archaeology [124]. Such an approach to

technology (including remote sensing and visualization methods) is consistent with Heidegger's view that the greatest threat of technology is the formation of attitudes in which we treat a given technology as a provider of ready-made solutions. The applied technology, however, obscures our reality and consequently, we ignore other alternative solutions, interpretations, or possibilities [63] (p. 13).

Another important issue in these reflections on aesthetics relates to conventionalization. Contemporary technology moves toward standardization, repetition of patterns, comparability, and homogeneity. In the practiced discourses "The fact must be expressed in the style of thought collective" [95] (p. 102). Thus, the generated visualizations are adjusted to stylistic conventions, in which (aesthetic) visualization itself becomes a standard in the "research" procedure, and its aesthetics become inscribed in culturally fixed patterns (but not necessarily in postulated rules of standardization of visual representations, e.g., [53,125]). In further consequence, the aesthetics of visualization may replace the reflection on the meaning of an object in the past and in contemporary discourse. This state of affairs leads to the question: Why do archaeologists approach contemporary visualizations so exaggeratedly and uncritically? In a sense, Heidegger provides the answer to this question—that science has stopped thinking because it focuses on the technical approach.

In the contemporary debate on digital humanities, Heidegger's position is interesting because it allows us to gain a distance to the technical approach to the humanities, and reveals the necessity of turning to the original thinking that underlies all thinking, including thinking about the humanities [126] (p. 137). This seems to be the essence of the matter. Thinking should be the basis for all activities of archaeologists. These activities cannot be limited to collecting and describing (also visualizing) data. Critical reflection and understanding cannot be limited only to *black boxes*, but to all components of the research procedure. Referring again to Heidegger, the lack of critical reflection on the created and used visualizations may lead to obscuring (past) reality and perpetuating the ideas proposed by archaeologists themselves and/or even by experts cooperating with them. And the use of once fixed images, additionally subject to cultural standardization, contributes to the spread of Heidegger's "idle talk" in the context of visual assertions (and through visualization) about reality (also past reality). According to Heidegger, "idle talk" is a superficial statement that only roughly refers to beings. Speech or text (or visualization in archaeology) does not touch beings as it is, but communicates through repetition. Such a statement quickly becomes popular, "authoritative," and inscribes itself in power relations losing its reference in being [18] (pp. 157–159). Additionally, the excessive pursuit of efficiency, resulting in the loss of the essence, contributes to the formation of an intellectual desert [126] (p. 144).

## 9. Final Remarks

In trying to answer the questions posed about the extent to which changes in technology (including remote-sensing methods) influence the understanding of the place of image/visualization in archaeological research and the cognitive status of images, it is worth remembering that it is not only the cultural environment that shapes archaeology, but also what archaeologists themselves think about archaeology and its cognitive capabilities (obviously affected by their own cultural contexts).

In 1973 David Clarke published an article *Archaeology: The Loss of Innocence* in which he summarized the changes in archaeology and its setting from the 1950s to the 1970s. Clarke wrote that "The loss of disciplinary innocence is the price of expanding consciousness; certainly the price is high but the loss is irreversible and the prize substantial. Although the loss of disciplinary innocence is a continuous process we can nevertheless distinguish significant thresholds in the transitions from consciousness through self-consciousness to critical self-consciousness" [127] (p. 6). One may ask whether the "loss of innocence" has affected all archaeologists, and whether it has covered all the necessary areas of critical self-consciousness. I have no doubt that in practice a number of remote sensing archaeologists remain innocent in the way Clarke puts it [128–135]. The place of technology (including remote sensing methods) and forms of visualization is only one of the fields in which traditional innocence manifests itself. It is a consequence of what Fleck describes as "thought styles" and "thought collectives," which in themselves inhibit critical reflection and influence the rejection of

different views, perpetuating the innocence. Technology (especially remote sensing methods) itself does not constitute progress in archaeology. First of all, archaeologists need critical reflection on what they do, how they do it, and why they do it [136]. Only with such an attitude can we eliminate the threat to culture posed by technology as understood by Heidegger.

There are many different questions raised in the text above and the readers may not find simple answers to them. Rather, it is my intention to ask questions, not to provide answers, especially unequivocal ones. Gone are the days when one way specific of thinking could be imposed. If these questions have encouraged the readers to try to find their own answers, I will consider it a success.

**Funding:** This research received no external funding

**Acknowledgments:** I have found writing this paper a challenge, as it has demanded that I address some unfamiliar topics—for this reason I would like to thank all those who supported me in this struggle. I have consulted philosophical and theoretical archaeology issues with Danuta Minta-Tworzowska, Anna Pałubicka and Henryk Mamzer as well as automatic detection topics with Jacek Marciniak. Their comments were very valuable and made me believe that I have not made any glaring mistakes in these fields. I would like to thank the editors of SI (Arianna Traviglia, Dave Cowley, and Geert Verhoeven) for their openness to accept such an unusual text and their continued support, even if we differed on some issues. I would also like to thank the anonymous peer reviewers for their insightful comments that allowed me to look at my text from a different angle. I did not accept all their comments, and this is certainly due to our different "thought style" and different scientific and cultural experiences. Of course, I am responsible for all narrative errors or inconsistencies.

**Conflicts of Interest:** The author declares no conflict of interest.

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
