# Peer review of "Power and/or Penury of Visualizations: Some Thoughts on Remote Sensing Data and Products in Archaeology"

_remotesensing, doi:10.3390/rs12182996_

Round 1

Reviewer 1 Report

This manuscript presents a series of philosophical issues organized around the idea that the products of remote sensing are cultural products in-and-of themselves and warrant critical reflection as such by archaeologists. Specifically, the author reviews a series of philosophical concepts pertinent to the production, analysis, and communication of different forms of imagery.

The author builds a case for the need for archaeologists to analyze themselves and their research and analytical processes as new remote sensing technologies are developed, modified, adopted, and integrated into practice. This is done by discussing a series of philosophical issues raised by a variety of philosophers.

Section 1 deals with issues of ontology, epistemology, and discourse, but is muddled by wordiness and poor flow. Section 2 effectively expands on issues of ontology and processes of creating representations. Section 3 addresses Heidegger’s conceptualization of technology in society, a useful perspective archaeologists should be reminded of. Section 4 suggests the use of phenomenological and hermeneutical approaches to the how archaeologists conceive of technology and its use. Section 5 presents a useful model for archaeologists to use when thinking about data acquisition in the context of culture. Section 6 draws on previous ideas to present an excellent model for understanding the technical production of the past. I don’t find that McLuhan’s tetradic model adds anything useful to the section though. It seems more like a distracting side-trip. The technical discussion of data classification is excellent and deserves expansion and concern by all archaeologists using such technologies (meaning almost all of them). Section 7 is a nice reminder of the implications of Fleck’s ideas for how archaeologists interpret the products of remote sensing. Section 8 usefully pulls apart the ways archaeologists use the products of visualization. The short discussion of the division between technician and archaeologist identifies a critical issue inhibiting the effective creation, use, and interpretation of visualizations. This topic deserves its own article.

Taken individually or as a whole, archaeologists would benefit from more explicit concern and application these ideas. Unfortunately, the raising of pertinent philosophical issues is intensely weakened by many mistakes in writing and argumentation.

The abstract, introduction, and section one are poorly written. The writing is wordy. The ideas are presented obliquely. They do not identify clearly the subjects of the rest of the manuscript or put the ideas into context for archaeology and remote sensing. The subsequent sections are much more coherent in organization and less wordy.

The greatest weakness of the manuscript is that it hinges on overgeneralization,
overstatement, and a superficial and artificial opponent. This is exacerbated by describing an oversimplified and generalized characterization of archaeological practice near the beginning in section 2. Subsequently, the author inserts numerous instances of personal claims of the commonality of this superficial view of archaeology and never attempts to substantiate it with evidence. I cannot find any attempt to substantiate these claims in reviewing the references.

Hodder and Binford had each other to use for their debate about archaeological thought and practice. The author of this manuscript only presents a simplistic image of pre post-processual archaeology and never attempts to substantiate or flesh out this straw man.

The use of such a superficial and artificial opponent is not even necessary. The ideas presented stand on their own as important for maintaining a field that is critically reflexive in the face of changing technologies. The ideas and questions presented are important/essential for archaeologists to grapple with as they adopt, transform and use rapidly changing remote sensing technologies.

Author Response

Dear Reviewer 1,

I am grateful to you for kindly giving your time to read the manuscript thoroughly and providing valuable comments and suggestions, positive and negative. I would like to acknowledge that your suggestions have helped me in refining the manuscript and making it more precise. I am sure both the text itself and the main message might be controversial, not accepted by all. But it might be its value – invitation to discussion.

Yours,

Włodzimierz Rączkowski

Reviewer 2 Report

The paper discusses and analyses the cognitive status of technology and visualization in the context of archeological research and poses questions regarding the interpretation of images. The author also refers to the ontological, epistemological and discursive perspectives of the new technological advances in the fields of remote sensing and archaeology.

The content is interesting and consisting but the contribution novelty is not a strength, since remote sensing techniques, and especially image-based methods for data collection and processing, have been evolved by documenting archaeological assets in an utterly new way (3D models, point clouds, orthophoto maps etc.) rather than with simple images. The dissemination of these products is also aligned with new trends and emerging technologies like online 3D visualization, virtual and augmented reality, semantics, BIM, 3D GIS etc. However, only virtual reality and some classification and features detection techniques are mentioned regarding the topic.

Although the subject matter of the paper suits the scope of the journal, I do not think that the way it is presented (first person point of view) is inline with the journal’s standards. The paper requires significant editing and major revisions need to be made. Overall, I recommend the paper to be published after major revisions (English language and content).

Here is a list of points that should be reconsidered:

Introduction:

-Row 44: The new digital technologies in the archaeological field are undefined. In section 3 the author elaborates on Virtual Reality and in section 6 on selected classification and detection (automatic, semi-automatic) methodologies that should be also stated here. Consider also to mention some of the remote sensing advances by referencing to previous works such as:

Opitz, R., Herrmann, J., 2018. Recent Trends and Long-standing Problems in Archaeological Remote Sensing. Journal of Computer Applications in Archaeology 1, 19–41. https://doi.org/10.5334/jcaa.11

Tapete, D., 2018. Remote Sensing and Geosciences for Archaeology. Geosciences 8, 41. https://doi.org/10.3390/geosciences8020041

Bezzi, L., Bezzi, A., Boscaro, C., Feistmantl, K., Gietl, R., Naponiello, G., Ottati, F., de Guzman, M., 2018. Commercial Archaeology and 3D Web Technologies. Journal of Field Archaeology 43, S45–S59. https://doi.org/10.1080/00934690.2018.1505410

Ferdani, D., Demetrescu, E., Cavalieri, M., Pace, G., Lenzi, S., 2020. 3D Modelling and Visualization in Field Archaeology. From Survey To Interpretation Of The Past Using Digital Technologies. Groma 15. https://doi.org/10.12977/groma26

-Row 63: It is not clearly mentioned which the problems are. The questions stated in the previous paragraph can be considered more than facts than issues need to be adressed. Therefore, the introduction lacks a clear problem statement and a contribution section.

Starting point: identification of research practice area:

Row 89: The sentence is incomprehensible. Consider to elaborate on the far-reaching research consequences in order to help the reader to understand the point.

Row 120 – 123: It is not clear what the author is trying to communicate here.

Technology and data visualization: an ontological perspective

Row 221: Consider of rephrasing the sentence.

Row 229: Consider of rephrasing the sentence.

Row 232: Photography is a form of visualization and representation. It is not defined which type of visualization method the author refers to.

Data acquisition:

Row 292: The statement is not illustrated by the provided data; the spectral bands of the satellites cannot be considered as a proof of concept in order to justify that each remote sensing method applies to a specific case study.

Row 484: The word “bias” is repeated many times in the following paragraphs but its meaning is unclear for the less expert reader. Consider providing an example of bias in archaeology.   

Conclusion:

The analysis results and/or the discussion are not clearly presented in this section; instead of this, the author is providing more references. Consider writing an evidence-based conclusion.

Author Response

Dear Reviewer 2,

I thank you for your comments, which have helped me to better enhance several aspects in the revised manuscript.

Best,

Włodzimierz Rączkowski

Reviewer 3 Report

The English is perfect. Only one formatting issue. Made some comments for author to consider and either accept or not before publication.

Author Response

Dear Reviewer 3,

I am very grateful to you for kindly giving your time to read the manuscript thoroughly, the acceptance of the text (in general) as well as providing several valuable comments and suggestions. I am sure both the text itself and the main message might be controversial, not accepted by all. But it might be its value as well as invitation to discussion.

Round 2

Reviewer 1 Report

I am surprised how the new draft of this manuscript is not revised in relation to the comments on made on the original draft.

I still believe that archaeology in general would benefit from discussing many of the ideas raised this manuscript. The philosophical issues surrounding treating products of remote sensing as cultural products warrant critical reflection as such by archaeologists. And archaeologists tend to be distracted from broader concerns by the nuts-and-bolts of practice and the demands of grant writing and publishing.

The revisions in this draft cleans up the text and clarifies discussions of some of the issues raised, but does not address the substantive problems.

Is still don't see why McLuhan's tetradic model is included. What does it add to the argument presented?

The greatest weakness of the manuscript continues to be the reliance on undocumented overgeneralization, overstatement, and a superficial and artificial opponent. The author still inserts many instances of personal claims of the commonality of this superficial view of archaeology and **never** attempts to substantiate it with evidence. I still cannot find any attempt to substantiate these claims in reviewing the references. What is left is an essay that presents one persons philosophical views on the nature of meaning of archaeological remote sensing productions presented in relation to a straw man opponent .

Once again, the use of such a superficial and artificial opponent is not necessary. The ideas presented stand on their own as important for maintaining a field that is critically reflexive in the face of changing technologies. The ideas and questions presented are important/essential for archaeologists to grapple with as they adopt, transform and use rapidly changing remote sensing technologies.

Author Response

Dear Reviewer 1,

again, thank you very much for your effort to improve the text.

Reviewer 2 Report

The author has revised the manuscript according to the majority of the provided suggestions. Specifically, (i) the abstract's revision better reflects the subject matter and aim of the paper, (ii) the addition of the references regarding the new digital technologies accentuates the evolution of the domain and elucidates the literature review of the corresponding section of the manuscript for non-specialists, (iii) the suggested sentences have been rephrased and, (iv) the modification of the title of "Conclusions" section better suits the style of the manuscript.

I respect the author's point but I still disagree on the use of "first person" form. I also insist on removing or replacing the Table 1 since the range of spectral bands is not a metric to decide which device is "better". Actually, factors like resolution and/or type of sensors produce different data types and determine the "quality" and applicability of each device. Consider to include such a comparison. 

The overall evaluation of the revised paper is quite positive and minor revisions should be made before publication. 

Author Response

Dear Reviewer 2,

Thank you very much for your effort to work on improving the paper.

This manuscript is a resubmission of an earlier submission. The following is a list of the peer review reports and author responses from that submission.